# A unifying modelling of multiple land degradation pathways in Europe

Remus Prăvălie[1,2,3] ✉, Pasquale Borrelli [4,5], Panos Panagos [6], Cristiano Ballabio [6], Emanuele Lugato [6], Adrian Chappell [7], Gonzalo Miguez-Macho [8], Federico Maggi [9], Jian Peng [10], Mihai Niculiță [11], Bogdan Roșca[12], Cristian Patriche[12], Monica Dumitrașcu[13], Georgeta Bandoc[1,3], Ion-Andrei Nita [14] & Marius-Victor Birsan [13]

Land degradation is a complex socio-environmental threat, which generally occurs as multiple concurrent pathways that remain largely unexplored in Europe. Here we present an unprecedented analysis of land multi-degradation in 40 continental countries, using twelve dataset-based processes that were modelled as land degradation convergence and combination pathways in Europe's agricultural (and arable) environments. Using a Land Multi-degradation Index, we find that up to 27%, 35% and 22% of continental agricultural (~2 million km²) and arable (~1.1 million km²) lands are currently threatened by one, two, and three drivers of degradation, while 10–11% of pan-European agricultural/arable landscapes are cumulatively affected by four and at least five concurrent processes. We also explore the complex pattern of spatially interacting processes, emphasizing the major combinations of land degradation pathways across continental and national boundaries. Our results will enable policymakers to develop knowledge-based strategies for land degradation mitigation and other critical European sustainable development goals.

Land, with its defining elements of soils, vegetation, and inland water resources[1,2], is threatened worldwide by numerous degradation processes that may drive the decrease or collapse of biological and economic productivity, biodiversity, ecological integrity and complexity, or ecosystem functions that support the delivery of primary ecosystem services[3,4]. There are multiple negative consequences of land degradation (LD), which have profound implications for agricultural productivity[5], food security[6], climate stability[7], environmental sustainability[8], and economic prosperity[1]. For the last case alone, it is estimated that the global economic impact of LD, assessed through loss of various ecosystem services, could total the astonishing cost of 6.3–10.6 trillion US dollars annually[1,9].

The multisectoral impact of this major environmental disturbance is profoundly influenced by the occurrence patterns of LD

[1]University of Bucharest, Faculty of Geography, 1 Nicolae Bălcescu Street, 010041 Bucharest, Romania. [2]University of Bucharest, Research, Institute of the University of Bucharest (ICUB), 90–92 Panduri Street, 050663 Bucharest, Romania. [3]Academy of Romanian Scientists, 54 Splaiul Independentei Street, 050094 Bucharest, Romania. [4]Department of Environmental Sciences, Environmental Geosciences, University of Basel, Basel, Switzerland. [5]Department of Science, Roma Tre University, Rome, Italy. [6]European Commission, Joint Research Centre (JRC), Ispra, Italy. [7]School of Earth and Environmental Sciences, Cardiff University, Wales, United Kingdom. [8]CRETUS, Non-Linear Physics Group, Faculty of Physics, Universidade de Santiago de Compostela, Galicia, Spain. [9]Environmental Engineering, School of Civil Engineering, The University of Sydney, Sydney, NSW, Australia. [10]Laboratory for Earth Surface Processes, Ministry of Education, College of Urban and Environmental Sciences, Peking University, Beijing, China. [11]Alexandru Ioan Cuza University, Faculty of Geography and Geology, Department of Geography, 20A Carol I Street, 700506 Iași, Romania. [12]Romanian Academy, Iași Divison, Geography Department, 8 Carol I Street, 700505 Iași, Romania. [13]Institute of Geography, Romanian Academy, 12 Dimitrie Racoviță Street, 023993 Bucharest, Romania. [14]VisualFlow, 140 Aurel Vlaicu, 020099 Bucharest, Romania. ✉e-mail: pravalie_remus@yahoo.com

processes. The spatial pattern of various LD pathways was explored in Europe and worldwide in many unidimensional studies, focused so far on the separate analysis of certain distinct processes, like water erosion[10,11], wind erosion[12,13], soil salinization[14,15], soil compaction[16,17] or soil organic carbon loss[18,19]. These examples and many other studies generally investigated LD from a unilateral perspective, in which the individual processes were examined as isolated and disconnected pathways of various existing processes[2] that usually interact in the LD mechanism.

Therefore, international scientific literature has generally ignored the multidimensional nature of LD, which by its definition involves a cumulative approach of various land degradative facets. Some well-known global specialized reports (which include Europe) assessed several LD processes, but often in an individual manner and focusing on soil degradation, which is only one of the three key elements that define the broader concept of land. Such representative scientific reports include the Global Assessment of Soil Degradation[20], the World Atlas of Desertification (first, second, and third editions)[3,21,22], the Global Assessment of Land Degradation and Improvement[23], the Assessment Report on Land Degradation and Restoration[4] and the Climate Change and Land Special Report[7].

There are however certain previous attempts to examine some multiple LD processes (land multi-degradation), in a synergistic approach. Such a rare example is a recent study that investigated the co-occurrence (simultaneous presence) of five major degradation processes (water erosion, soil salinization, soil organic carbon loss, vegetation degradation, and aridity) for detecting land multi-degradation in global arable landscapes[6]. The shortcoming of this previous assessment is however the lack of integration of other key processes in modelling land multi-degradation. Another assessment of LD in a comprehensive manner is found in the World Atlas of Desertification Report (third edition), which attempted to assess both causes and some processes involved in worldwide LD, through a "convergence of evidence" approach[3]. Recently, the same approach was applied across the European Union (EU)[24].

However, some significant limitations still persist in these analyses, such as omitting many important specific processes and the mixture of various biophysical and socio-economic drivers of LD, the coincidence (or convergence) of which does not concretely (directly) indicate this disturbance, but rather the fact this environmental issue may exist[3,24]. While other studies explored the complex nature of LD based on certain multicriteria susceptibility models applied at global[25] or European[26] scales, they too generated results that did not focus on actual LD processes, but on examining the synergic effects of various ecological and socio-economic conditions potentially leading to LD.

To address the complex issue of land multi-degradation, a multi-process modelling approach of LD is essential in Europe. This approach can be crucial for applying various agricultural (e.g. Common Agricultural Policy)[27], climate (the European Green Deal)[28], and sustainable development (Sustainable Development Goals – SDGs)[11] policies on the continent. Consequently, a comprehensive analysis of LD can define a powerful decision-making support tool for European and national policies designed to mitigate LD hotspots and support food security, climate stability, and environmental sustainability throughout the continent.

In this study, we model, quantify, and map the convergence pattern of twelve LD processes in Europe using the latest state-of-the-art datasets. Our entire analysis is focused on continental (pan-European) agricultural environments, which are critically important for food production, but generally highly vulnerable to multi-degradation. Essentially, our research integrates a complex set of LD processes that are strategically important to continental agricultural productivity, thus trying to provide a solid scientific basis for a more realistic and efficient implementation of LD-related policies across Europe.

## Results and Discussion

### The continental picture of individual land degradation processes

We used a large set of geospatial data including twelve LD processes (water erosion, wind erosion, soil organic carbon loss, soil salinization, soil acidification, soil compaction, soil nutrient imbalances, soil pollution via pesticides, soil pollution via heavy metals, vegetation degradation, groundwater decline, aridity), which are highly representative for agricultural productivity and that were collected from various sources ($n = 6$) or developed in this study ($n = 6$) (see Methods). An overview of individual drivers of degradation is provided in Fig. 1, which provides their essential spatial characteristics before examining the continental status of multiple converging (co-occurring) processes.

The results showed different patterns of LD processes across Europe, which were examined and mapped according to some specific critical thresholds (classes), documented in the literature for each driver of degradation (see Methods). Our findings revealed that soil pollution via pesticides has, surprisingly, the largest spatial footprint at continental level (52% of the cumulated agricultural area of the 40 investigated countries, or ~1.10 million (M) km$^2$), of all analysed processes (Fig. 1h). This process, the vast spatial extent of which is explained by the very large number of substances (dozens of herbicides, insecticides and fungicides) considered in the modelled data of pesticide risk[29], is followed by soil nutrient imbalances (39% or ~0.82 M km$^2$) (Fig. 1g), soil pollution via heavy metals (31%, ~0.60 M km$^2$) (Fig. 1i), and aridity (26%, ~0.54 M km$^2$) (Fig. 1l). These four LD processes can be considered the most important in terms of spatial footprint, keeping in mind that each process affects over a quarter of European agriculture.

In contrast, soil salinization (1% or ~0.02 M km$^2$) (Fig. 1d), vegetation degradation (3%, ~0.07 M km$^2$) (Fig. 1j), groundwater decline (4%, ~0.08 M km$^2$) (Fig. 1k) and wind erosion (5%, ~0.10 M km$^2$) (Fig. 1b) cover the smallest agricultural areas in Europe. Thus, according to the percentage data, these 4 processes seem to be the smallest threats to land productivity, even though their absolute spatial footprint is still notable. While country-level statistics are far more diverse, some large countries with vast agricultural areas affected (up to >50% or even >75% of national agricultural lands) by critical conditions of soil pollution via pesticides (e.g. Poland, Italy or Spain) (Fig. 1h) and heavy metals (France, Italy or Greece) (Fig. 1i), or of soil nutrient imbalances (UK or Germany) (Fig. 1g) and aridity (Romania or Spain) (Fig. 1l), is truly remarkable. Other important LD hotspots (>25% or even >50% of agricultural lands classified as "Critical") can be observed sporadically especially for water erosion (countries from the Mediterranean and Balkan regions) (Fig. 1a), soil compaction (e.g. Baltic countries) (Fig. 1f), and soil acidification (Nordic countries) (Fig. 1e).

### Land multi-degradation pattern in Europe

By fusing the twelve geospatial databases, we created Land Multi-degradation Index (LMI) (see Methods), which highlights the number of interacting processes throughout the continent (Fig. 2). LMI revealed between one and ten co-occurring processes in Europe, which were grouped into five degradation classes (the "No degradation" class was approached separately, as it entails the absence of degradation conditions) – very low (1 process identified at pixel level), low (co-occurrence of 2 LD processes), medium (3), high (4) and very high (≥5) (Fig. 2), considering certain key statistical criteria and the reasoning of an easy interpretation of LMI results (see Methods). This classification was explored across the entire European agricultural land area (Fig. 2a), and a special focus was given to the arable lands (Fig. 2d), which are highly relevant for ensuring crop production and food security in Europe.

Considering LMI classes 1–3, it can be noted that large parts of the two land use categories are exposed to one (up to 27% of the European

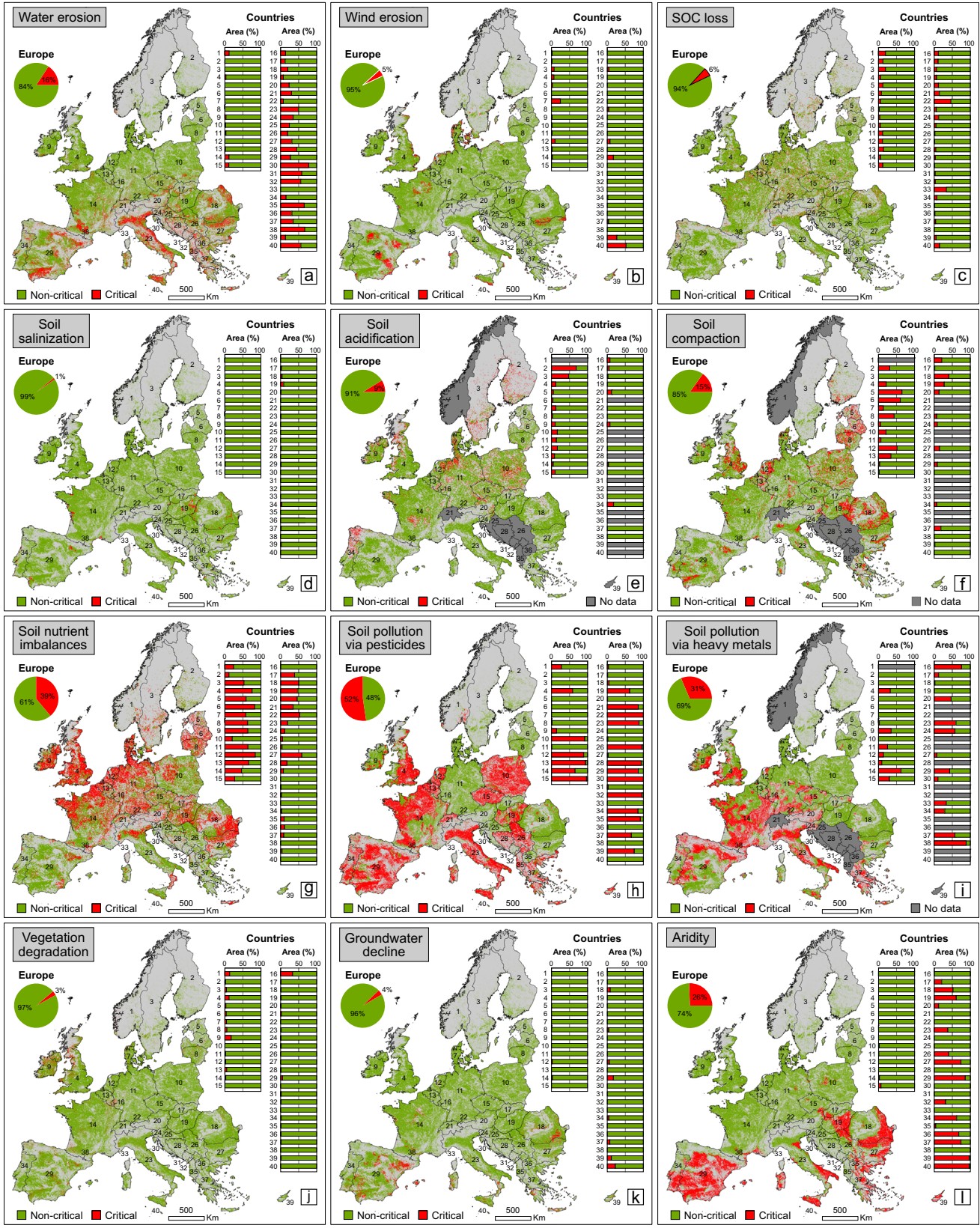

agricultural/arable area), two (up to 35%), and three (22%) drivers of degradation (Table 1). These first three classes highlight a very low, low, and medium vulnerability of lands to degradation, and total over 80% of the areas investigated in this study (Table 1). The next two classes are spatially limited, but significantly more important by reflecting the highest intensity of LD. For this reason, LMI classes 4 and

5 should be explored more closely throughout the pan-European agricultural and arable lands.

It was found that about a tenth of European agricultural lands are affected by high (8% of agricultural area or ~0.16 M km²) and very high (2%, ~0.04 M km²) degradation (Table 1), especially in the continent's southern, north-western and central south-eastern regions

**Fig. 1 | The continental spatial pattern of twelve LD processes approached separately in Europe. a** Water erosion. **b** Wind erosion. **c** Soil organic carbon (SOC) loss. **d** Soil salinization. **e** Soil acidification. **f** Soil compaction. **g** Soil nutrient imbalances. **h** Soil pollution via pesticides. **i** Soil pollution via heavy metals. **j** Vegetation degradation. **k** Groundwater decline. **l** Aridity. The "Critical" class was delimited for each process as per the details featured in the Methods section. Horizontal columns represent percentage-based areas of land degradation classes ("Critical" and "Non-critical"), related to the absolute area of national agricultural lands (which can be found in the Supplementary Information section). Light grey highlights non-agricultural lands, while dark grey indicates the countries with no data available for the specific processes. The numbers inside the maps and diagrams are the investigated countries, ordered (from north to south) in descending order considering the maximum latitude values of their northern limits. The complete list of the 40 countries is featured in the Supplementary Tables. The source data for the graphs in this figure are provided as a Source Data file.

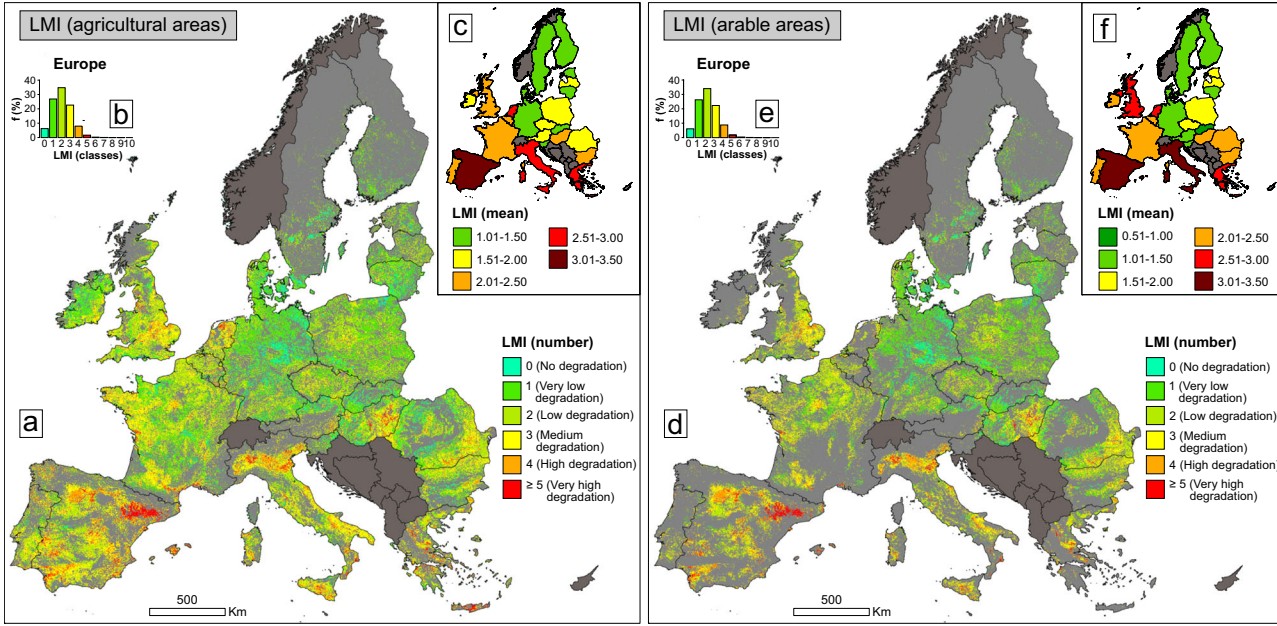

**Fig. 2 | Spatial pattern of land multi-degradation in Europe. a** Spatial distribution of LMI values (number of co-occurring processes) in agricultural landscapes. **b** Histogram of LMI values for European agricultural lands. **c** Average number of co-occurring processes in agricultural environments of continental countries. **d** Spatial distribution of LMI values (number of co-occurring processes) in arable landscapes. **e** Histogram of LMI values for European arable lands. **f** Average number of co-occurring processes in arable environments of continental countries. LMI is the acronym for Land Multi-degradation Index. Light grey highlights non-agricultural/non-arable lands, while dark grey indicates the (masked) countries with incomplete data (9–10 input layers out of 12) for LMI modelling. The source data for the graphs in this figure are provided as a Source Data file.

**Table 1 | Spatial extent (in km² and %) of LMI classes in agricultural/arable environments of Europe**

| No. | LMI classes (number of co-occurring processes) | Agricultural lands | | Arable lands | |
|---|---|---|---|---|---|
| | | km² | % | km² | % |
| 1 | No degradation (0)[a] | 120,963 (±29,031) | 6.16 (±1.48) | 66,410 (±15,938) | 6.07 (±1.46) |
| 2 | Very low degradation (1) | 523,824 (±167,624) | 26.70 (±8.54) | 287,879 (±92,121) | 26.33 (±8.43) |
| 3 | Low degradation (2) | 678,224 (±149,485) | 34.57 (±7.62) | 372,937 (±82,198) | 34.11 (±7.52) |
| 4 | Medium degradation (3) | 440,473 (±66,655) | 22.45 (±3.40) | 244,081 (±36,936) | 22.32 (±3.38) |
| 5 | High degradation (4) | 155,265 (±25,020) | 7.91 (±1.28) | 94,654 (±15,253) | 8.66 (±1.39) |
| 6 | Very high degradation (≥5)[b] | 43,352 (±8,919) | 2.21 (±0.45) | 27,460 (±5,650) | 2.51 (±0.52) |

[a]agricultural/arable lands unaffected by degradation processes. [b]most frequently five concurrent processes, according to the LMI histograms for agricultural/arable areas (Fig. 2b, e). % – the percentage-based area of the number of convergent processes (0, 1, 2, 3, 4, ≥5), related to the absolute area of continental agricultural (1,962,101 km²)/arable (1,093,421 km²) lands. The values in parentheses (±) are error ranges obtained by applying a Random Forest classification model (see Methods). All these European statistics were extracted after excluding the countries with incomplete data for LMI modelling (Fig. 2).

(Fig. 2a). While the countrywide picture is more diverse, the most important hotspots in terms of percentages of concurrent processes (LMI classes 4 and 5 combined) are Spain (~30% of the national agricultural area), Greece (26%), Italy (23%), Netherlands (20%) and Hungary (14%) (Supplementary Table 2, Fig. 2a). Also, Spain (~0.07 M km²), Italy (~0.04 M km²), France (~0.02 M km²), UK (~0.01 M km²) and Greece (~0.01 M km²) are the most remarkable in terms of absolute areas affected by high and very high degradation (which can be determined based on percentages and absolute agricultural areas featured in Supplementary Table 2). All seven countries are among the most threatened by multi-degradation, also considering the national average of co-occurring processes (Fig. 2c). In

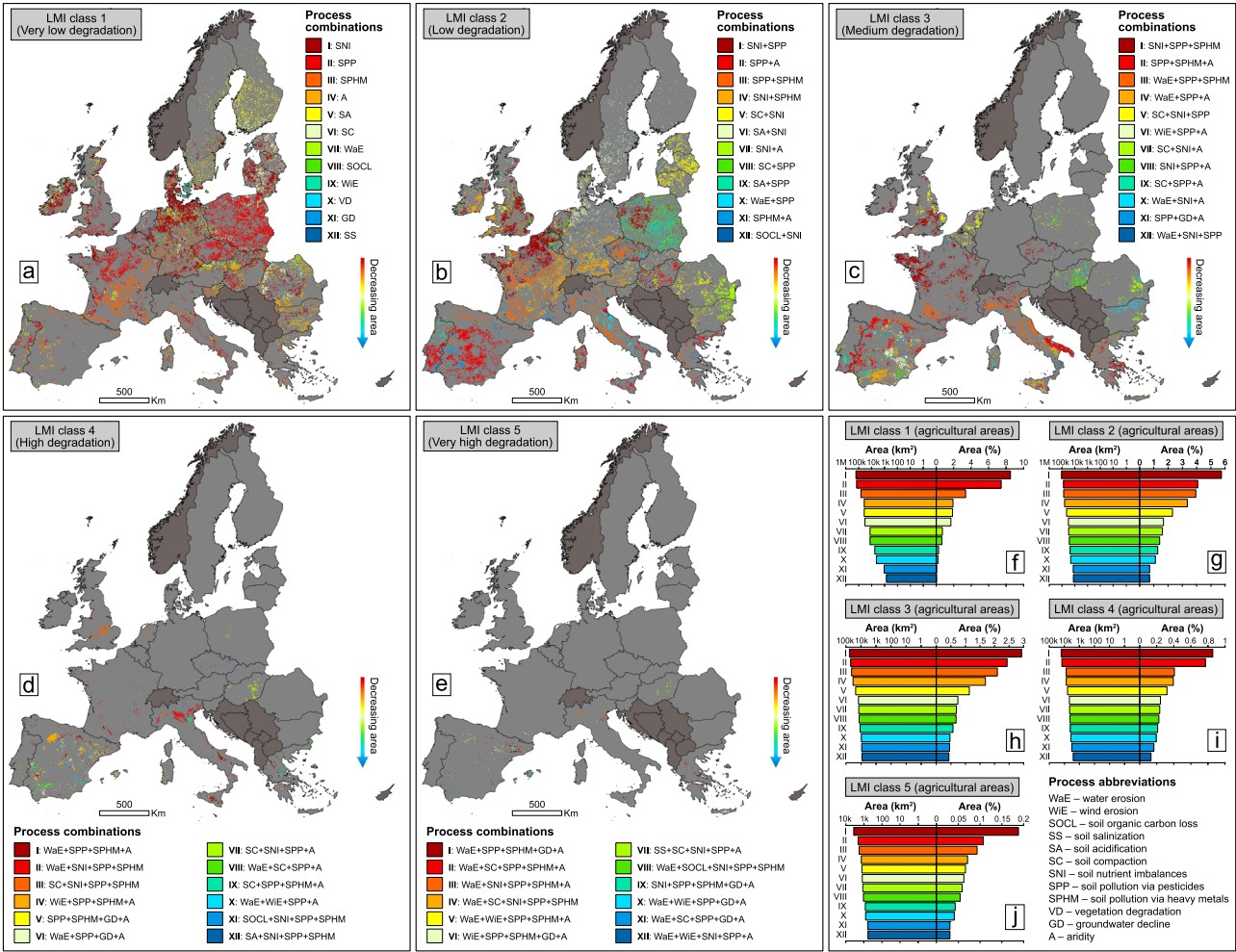

**Fig. 3 | Spatial pattern of interacting convergent processes in agricultural environments of Europe.** Spatial distribution of the dominant (most frequent) co-occurring process types (combinations) in LMI agricultural classes 1 (**a**), 2 (**b**), 3 (**c**), 4 (**d**) and 5 (**e**). Absolute and percentage-based (% of the total continental agricultural lands) spatial footprint of the dominant co-occurring process types in LMI agricultural classes 1 (**f**), 2 (**g**), 3 (**h**), 4 (**i**), and 5 (**j**). LMI is the acronym for Land Multi-degradation Index. In order to simplify the mapping of process combinations (which are very numerous for each LMI class, except for class 1), in this figure the twelve most important types of co-occurring processes in Europe were selected,

which, in terms of area, cumulatively account for at least 50% of all LMI class combinations. To better highlight the mapped process combinations, pixel size was increased to 5 km × 5 km, but the quantification of process combination areas (in km² and %) was done using the original data resolution of 500 m × 500 m. The countries in dark grey were masked as they contained incomplete data (9–10 input layers out of 12) for LMI modelling. The European statistics (**f**–**j**) were extracted after excluding the countries with incomplete data for LMI computation. The source data for the graphs in this figure are provided as a Source Data file.

contrast, Germany is the least affected (alongside several other European states), as it holds extensive lands (>0.03 M km²) with high agro-ecological potential (the "No degradation" class), which are not under the incidence of even a single LD process (Supplementary Table 2, Fig. 2a,c).

Also, large portions of Europe's arable lands are under the pressure of the most critical conditions, framed in LMI classes 4 (9%, -0.09 M km²) and 5 (3%, -0.03 M km²) (Table 1, Fig. 2d). Spain (36%), Italy (32%), Greece (27%), Hungary (14%) and Netherlands (14%) remain major hotspots of LMI classes 4 and 5 (combined) in terms of percentages, while the top 5 countries affected in terms of absolute areas consists of Spain (-0.05 M km²), Italy (-0.03 M km²), France (-0.01 M km²), UK (-0.01 M km²) and Romania (-0.01 M km²) (Supplementary Table 3, Fig. 2d). These states are also the most vulnerable in terms of the mean number of concurrent processes in national arable lands (Fig. 2f). At the opposite pole, Germany (-0.02 M km²) and Sweden (-0.01 M km²) hold the most extensive lands unaffected by one or more LD processes (Supplementary Table 3, Fig. 2d,f).

In order to statistically consolidate all these LMI results, we explored the potential uncertainties of our modelling, which may primarily result from defining the critical thresholds of multiple LD processes (see Methods). Thus, the possible uncertainties associated to the threshold-driven LMI values were quantified and mapped across Europe, using a Random Forest classification model (see Methods).

The results emphasized an overall prediction error of 17.6% (out-of-bag error) throughout the continent. Essentially, the uncertainties were defined as the probability that each modelled pixel falls in one of the LMI classes, according to Supplementary Fig. 1. The findings on the uncertainty (and sensitivity) analysis allowed us to define the geographical variability (Supplementary Fig. 1) of the LMI classes (associated to thresholds different from those that were defined according to scientific and policy related literature, as detailed in Methods), but also to estimate the prediction error associated to each LMI class area resulted in this study (Table 1).

Acknowledging some degree of uncertainty (Table 1, Supplementary Fig. 1), our multi-degradation approach becomes a better tool that can be highly useful for various EU policies (see Policy

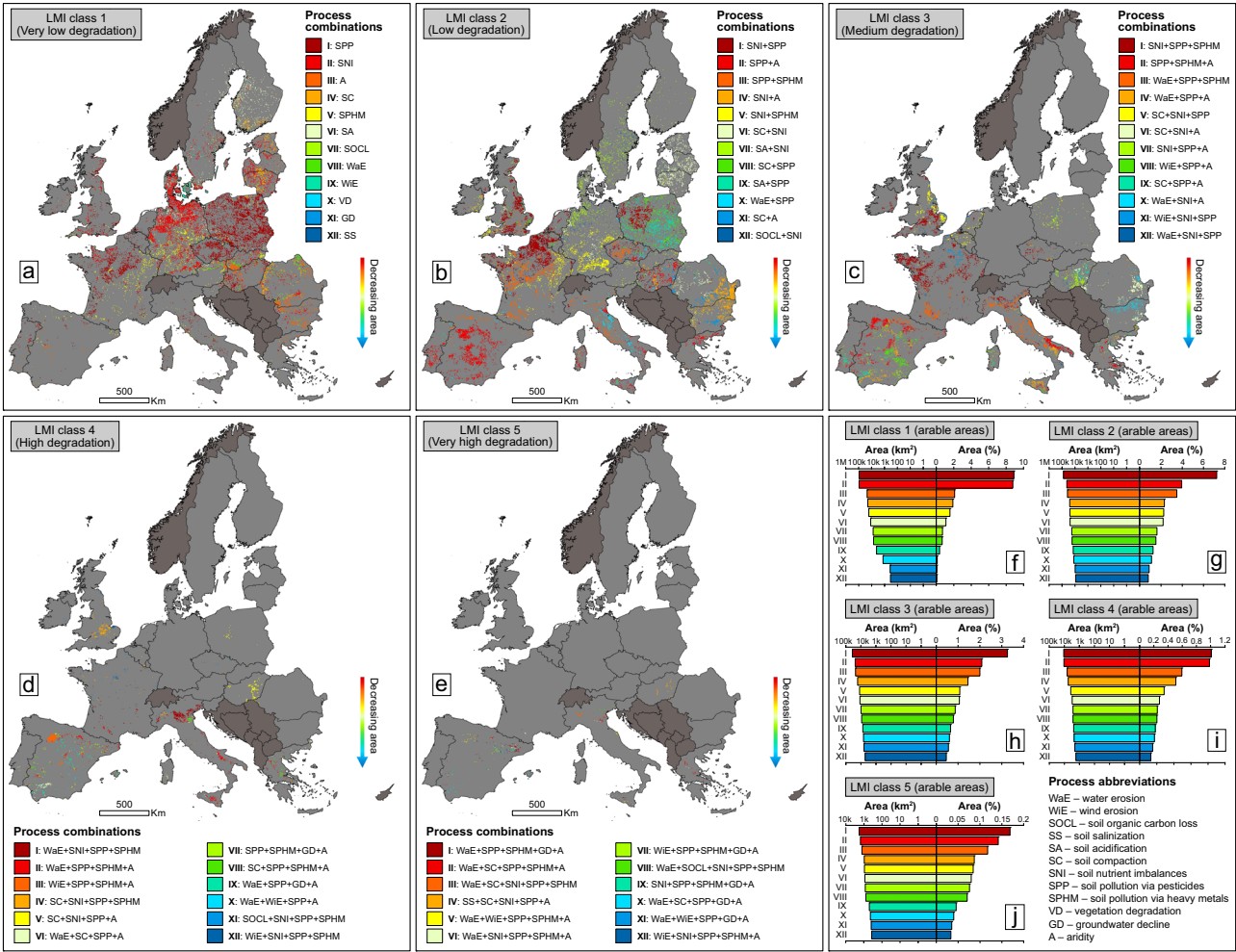

**Fig. 4 | Spatial pattern of interacting convergent processes in arable environments of Europe.** Spatial distribution of the dominant (most frequent) co-occurring process types (combinations) in LMI arable classes 1 (**a**), 2 (**b**), 3 (**c**), 4 (**d**) and 5 (**e**). Absolute and percentage-based (% of the total continental arable lands) spatial footprint of the dominant co-occurring process types in LMI classes 1 (**f**), 2 (**g**), 3 (**h**), 4 (**i**) and 5 (**j**). LMI is the acronym for Land Multi-degradation Index. In order to simplify the mapping of process combinations (which are very numerous for each LMI class, except for class 1), in this figure the twelve most important types of co-occurring processes in Europe were selected, which, in terms of area,

cumulatively account for at least 50% of all LMI class combinations. To better highlight the mapped process combinations, pixel size was increased to 5 km × 5 km, but the quantification of process combination areas (in km² and %) was done using the original data resolution of 500 m × 500 m. The countries in dark grey were masked as they contain incomplete data (9–10 input layers out of 12) for LMI modelling. The European statistics (**f–j**) were extracted after excluding the countries with incomplete data for LMI computation. The source data for the graphs in this figure are provided as a Source Data file.

implications). Scientifically, the uncertainty modelling framework helps moving closer towards robust, repeatable, and open data science to communicate with adjacent disciplines and better deal with complex LD challenges.

## Types of spatially interacting processes across the continent

In addition to the number of convergent processes, emphasized by LMI, the combination of concurrent processes is also relevant at the continental/national scale. Following the twelve dominant (most frequent) process combinations for each LMI class, we detected a complex pattern of interacting LD pathways, which are dominant in Europe's agricultural (Fig. 3) and arable (Fig. 4) landscapes.

The first three classes are marked by large-scale combinations of LD processes (LMI class 1 indicates, in fact, a single driver of degradation, with no other concurrent processes), which affects extensive agricultural areas continentally (Fig. 3a–c, f–h) and nationally (see detailed statistics in Supplementary Tables 4–6). Smaller, yet still remarkable areas can also be noticed for combinations of LMI classes 1–3 for continental (Fig. 4a–c, f–h) and national (Supplementary

Tables 9–11) arable lands. While the last two classes are significantly smaller, they are the most important due to the complex spatial interaction of four and five drivers of degradation.

For agricultural lands, we found the most common four concurrent processes (LMI class 4) represented by water erosion (WaE), soil pollution via pesticides (SPP), soil pollution via heavy metals (SPHM) and aridity (A) (the abbreviated combination WaE + SPP + SPHM + A), and water erosion, soil nutrient imbalances (SNI), soil pollution via pesticides and via heavy metals (WaE + SNI + SPP + SPHM) (Fig. 3d). Each combination exceeds 0.01 M km² (or >10,000 km²) (Fig. 3i) and affects Italy the most (Fig. 3d), with >10% of its agricultural lands marked by the two combinations of interacting processes (Supplementary Table 7). Considering all twelve major types of combinations, Spain is another epicentre of land multi-degradation (Fig. 3d), with ~13% of its agricultural lands affected by various associations of four LD processes (Supplementary Table 7).

We also detected twelve main associations of five concurrent processes (LMI class 5) (Fig. 3e), which in most cases exceed 1000 km² (Fig. 3j). The most extensive association totals almost 4000 km² (water

erosion, soil pollution via pesticides, via heavy metals, groundwater decline (GD) and aridity, or WaE + SPP + SPHM + GD + A) (Fig. 3j), mainly affecting north-eastern Spain (Fig. 3e). Generally, Spain, Italy and Greece are the main hotspots of the twelve types of combinations belonging to LMI class 5 (Fig. 3e, Supplementary Table 8).

A similar spatial pattern can also be observed for arable lands (Fig. 4d, e), with some differences in the area hierarchy of combined processes (Fig. 4i,j). We found two prominent cases (>10,000 km$^2$) with four concurrent processes (LMI class 4) consisting of water erosion, soil nutrient imbalances, soil pollution via pesticides and via heavy metals (WaE + SNI + SPP + SPHM), and water erosion, soil pollution via pesticides, via heavy metals and aridity (WaE + SPP + SPHM + A), respectively (Fig. 4d,i). Once more, Italy is the epicentre of the two major combinations (which affect 16% of its arable lands) (Supplementary Table 12), while Spain remains a four-process combination diversity hotspot (Fig. 4d). Also, while there are combinations of five convergent processes (LMI class 5) mainly in Mediterranean countries, they cover significantly smaller portions of arable lands (Fig. 4e,j, Supplementary Table 13).

## Policy implications

The United Nations (UN) General Assembly resolution A/RES/73/284 proclaimed 2021–2030 the UN Decade on Ecosystem Restoration, which has the aim to "prevent, halt and reverse the degradation of ecosystems worldwide". The SDGs as part of the 2030 UN agenda provide a roadmap for a sustainable world including 17 goals that are addressed with subsequent targets. 'Land Degradation Neutrality' (target 15.3) aims to achieve LD neutrality by 2030. Accordingly, the lack of information about the possible co-occurrence of different degradation processes at a global (or European) scale can represent a concrete limitation for the achievement of multiple targets of the UN agenda.

In Europe, in 2018 the European Court of Auditors, one of the seven institutions of the EU, assessed LD and desertification on a large continental scale. In a special report, the Court of Auditors recommended to the European Commission to better address LD and desertification in the EU, emphasizing the need to enhance the EU legal framework for soil and propose actions towards delivering the commitment made by the EU and the Member States to achieve LD neutrality in the EU by 2030. According to the recent report of the Mission 'A Soil Deal for Europe', it was estimated that 60–70% of all soils in the EU are unhealthy due to current management practices, pollution, urbanisation and the effects of climate change. The EU has placed the need for healthy soils at the core of the European Green Deal to achieve climate neutrality, zero pollution, sustainable food provision, and a resilient environment.

In this context, the European Commission has proposed the Soil Strategy for 2030 with specific actions in relation to climate change mitigation, circular economy, biodiversity, desertification, soil restoration, soil monitoring, and citizen engagement to enable the transition to healthy soils. The recently established EU Soil Observatory supports the implementation of the EU Soil Strategy 2030 and other relevant EU policies, such as the Common Agricultural Policy, the Zero Pollution Action Plan and the Farm to Fork Strategy. A more legally binding framework is also planned with the Soil Health Law that will contribute to the achievement of the Soil Strategy 2030 objectives and grant soils the same level of protection as water and air. However, the lack of knowledge related to the co-occurrence of different LD processes is a limitation to the EU targerts. In a broader context, this limitation remains an important obstacle to achieving the global LD targets of the UN agenda.

Our approach provides solid results relevant for the Soil Mission assessment and develops a comprehensive pan-European baseline for assessing land multi-degradation in Europe. Acknowledging some potential methodological limitations and uncertainties (see Methods), we contend that the set of evidences reported here serves as a basis for informing targeted LD mitigation strategies under the EU policies. Under these circumstances, we call on the European Commission to consider our complex findings on co-occurring and interacting LD processes in Europe, in order to more effectively stabilize and mitigate land multi-degradation, and ultimately to achieve a land degradation-neutral continent in the coming years.

Also, through the consistent and unprecedented examination of multiple drivers of degradation across 40 countries, our work presents broad perspectives beyond European policies. Scientifically, our modelling framework can represent a viable instrument to communicate with adjacent disciplines and move toward further integrated assessments among the soil-land-water nexus. Therefore, we consider the LMI proposed here can be a valuable interdisciplinary tool for the complex scientific assessment of LD, which is applicable in other regions of the planet if multiple and optimal environmental data are available.

## Methods

### Study area

This European analysis includes 40 continental states, 27 of which are member states of the EU. The study area comprises almost all European countries, except for seven (Vatican, Iceland, Belarus, Ukraine, Republic of Moldova, and transcontinental states Russia and Turkey) that were not included in the study, due to small size (Vatican, without agricultural areas) or lack of geospatial data for most of the analysed land degradation processes (in the remaining six countries). The 40 countries total an area of ~5 million (M) km$^2$ (approximately half of the European continent) and hold a combined agricultural area of ~2.10 M km$^2$ (~42% of the total area), over half of which (~1.14 M km$^2$) consists of arable areas.

### Data selection

In order to investigate land multi-degradation in Europe, we selected twelve processes that are highly representative for agricultural environments (Table 2). The twelve processes are the most relevant for highlighting the agricultural landscapes' degradation in Europe (and worldwide)[2], considering certain bio-physical mechanisms, general or particular, that trigger various negative effects in land productivity (Table 2). Several suggestive examples of such disruptive effects, which lead to the decrease or loss of land agro-ecological productivity, were documented for each degradation process, based on specialised literature (Table 2).

### Data acquisition/preparation

For the selected processes, we acquired/processed twelve geospatial databases. Essentially, we collected databases that were already available in their final form for six LD processes, the detailed processing information of which can be found directly in data sources – water erosion[11], soil organic carbon loss[19], soil salinization[30], soil acidification[31], soil compaction[32] and soil pollution via pesticides[29] (Table 3). For the other layers, we used various pre-existent data from other data sources, in order to refine (wind erosion) or model/obtain (soil nutrient imbalances, soil pollution via heavy metals, vegetation degradation, groundwater decline and aridity) the final data for the remaining six processes (Table 3).

Wind erosion was predicted by physically-based modelling of the aeolian sediment transport ($Q$) for a given particle size. The $Q$ is controlled mainly by two key properties: the momentum of the wind and the wind friction, influenced mainly by vegetation, which reduces the momentum reaching the unsheltered soil surface. The soil surface characteristics limit the entrainment of dry, loose, and available sediment, resulting in a specific threshold. That threshold is increased by soil moisture, which inhibits entrainment and for which a function is available. The availibility of sediment is limited by biogeochemical soil crusts/seals, but no parameterisation are currently available. Consequently, sediment is assumed to have an infinite supply and $Q$ is limited

**Table 2 | Various degradation processes (pathways) considered for modelling the agricultural land multi-degradation in Europe**

| No. | Land degradation processes/pathways | Examples of negative effects on agricultural land productivity |
|---|---|---|
| 1 | Water erosion | Degrading soil structure, reducing soil depth, or decreasing/losing the soil nutrient content[64–66] |
| 2 | Wind erosion | Accelerating dust emission, damaging crops by abrasion or reducing the organic matter content[8,12,13] |
| 3 | Soil organic carbon loss | Disrupting structural stability and water holding capacity of soils or decreasing soil fertility[19,67,68] |
| 4 | Soil salinization | Limiting plant growth due to phytotoxicity, water uptake difficulty, or soil organic carbon losses[69–71] |
| 5 | Soil acidification | Threatening soil bacterial diversity, increasing toxicity for plants, or limiting soil nutrient availability[72–74] |
| 6 | Soil compaction | Reducing soil porosity, shrinking oxygen and water supply to plants, or restricting root penetration[75–77] |
| 7 | Soil nutrient imbalances | Amplifying acidity and micronutrient deficiencies in soils, due to N or P excess, or limiting plant growth, due to N or P deficit[78–80] |
| 8 | Soil pollution via pesticides | Exerting stress on soil health via toxicity and decline in microbial community or earthworm activity[29,81,82] |
| 9 | Soil pollution via heavy metals | Poisoning the soil, injuring plants via chlorosis and necrosis, or hindering root growth and crop yields[38,39,83] |
| 10 | Vegetation degradation | Decreasing soil organic carbon via lower input to soil or through increased land exposure to water and wind erosion[2,84,85] |
| 11 | Groundwater decline | Depleting groundwater resources, inducing soil water stress, or inhibiting plant development[86–88] |
| 12 | Aridity | Generating surface low water availability and constant soil water deficit or triggering desertification[2,6,53] |

In this approach, the "land" concept includes soils, vegetation, and inland water resources, which is why the selected processes address the degradation of the three constituent components/ systems of lands. The first nine degradation processes (1–9) mainly affect the soil component, while the last three affect the vegetation (10) and water resources (11, 12).

**Table 3 | Characteristics of land degradation data used for modelling the agricultural land multi-degradation in Europe**

| No. | Land degradation data | Original resolution[a] | Time period | Metric/Unit of measure | Critical threshold/class[b] | Data source[c] |
|---|---|---|---|---|---|---|
| 1 | Water erosion | 250 × 250 m | 2012 | t ha$^{-1}$ yr$^{-1}$ | > 2 t ha$^{-1}$ yr$^{-1}$ | 11 |
| 2 | Wind erosion | 500 × 500 m | 2001–2021 | t ha$^{-1}$ yr$^{-1}$ | > 2 t ha$^{-1}$ yr$^{-1}$ | 13 |
| 3 | Soil organic carbon loss | 1 × 1 km | 2001–2015 | t C km$^2$ yr$^{-1}$ | <−0.1 t C km$^2$ yr$^{-1d}$ | 19 |
| 4 | Soil salinization | 1 × 1 km | 2008 | % | > 50 %[e] | 30 |
| 5 | Soil acidification | 500 × 500 m | 2019 | pH units | < 5.5 | 31 |
| 6 | Soil compaction | 1 × 1 km | 2008 | Susceptibility | H and VH SC | 32 |
| 7 | Soil nutrient imbalances | 1 × 1 km[f] 100 ×100 m[g] | 2010–2019 | kg/ha (N) mg/kg (P) | > 50 kg/ha/NUE > 0.9[h] > 50 mg/kg/< 25 mg/kg[i] | 31,33–35,89 |
| 8 | Soil pollution via pesticides | 10 × 10 km | 2015 | Risk score | H and VH RSC[j] | 29 |
| 9 | Soil pollution via heavy metals | 1 × 1 km[k] 500 × 500 m[l] 250 × 250 m[m] | 2009 | mg/kg | > 5 (As), > 1 (Cd), > 100 (Cr), > 20 (Co), > 60 (Pb), > 2 (Sb), > 50 (Ni), > 100 (Cu), > 0.5 (Hg)[n] | 38–40 |
| 10 | Vegetation degradation | 500 × 500 m | 2000–2015 | NDVI units | <−0.001 units yr$^{-1o}$ | 43 |
| 11 | Groundwater decline | 1 × 1 km | 2004–2013 | GTD (m yr$^{-1}$) | <−0.001 m yr$^{-1p}$ | 49 |
| 12 | Aridity | 1 × 1 km | 1981–2018 | AI (mm/mm) | < 0.65 mm/mm | 51,52 |

*m* meter, *km* kilometer, *ha* hectare, *t* ton, *C* carbon, *H* high, *VH* very high, *SC* susceptibility classes, *N* nitrogen, *P* phosphorous, *NUE* Nitrogen Use Efficiency, *RSC* risk score classes, *NDVI* Normalized Difference Vegetation Index, *GTD* Groundwater Table Depth, *AI* Aridity Index. [a]spatial resolution of the originally collected data, which were processed in this study at a common resolution of 500 × 500 m. [b]critical thresholds over/under which each land degradation process triggers the reduction or loss of agricultural land productivity. These thresholds used for modelling agricultural land multi-degradation were documented and set for each process based on scientific literature: water and wind erosion[10,61,90], soil organic carbon loss[2,6], soil salinization[30], soil acidification[4,72], soil compaction[32], soil nutrient imbalances[33,36,37,89], soil pollution via pesticides[29], soil pollution via heavy metals[38,42] and aridity[2,53]. As no concrete thresholds were found in the literature for vegetation degradation and groundwater decline, in these two cases some critical thresholds/classes were set in accordance with the reasoning explained in o and p. [c]source for databases that were already available and directly collected (processes with no. 1, 3, 4, 5, 6 and 8) or for pre-existing data used for refining/modelling/obtaining the other processes in this study (2, 7, 9, 10, 11 and 12). [d]negative statistically significant trends of soil organic carbon stock, detected during 2001–2015 using the MK test and *Sen's slope* estimator. [e]percentage of areas affected by saline and sodic soils (mainly Solonchaks and Solonetz). [f]for N data. [g]for P data. [h]for highlighting the N excess (>50 kg/ha) and deficit (NUE > 0.9) in soil. [i]for highlighting the P excess (>50 mg/kg) and deficit (<25 mg/kg) in soil. [j]high (3 < RS ≤ 4) and very high (RS > 4) risk of soil pollution (with dozens of pesticides in Europe), according to the data source. [k]for Arsenic (As), Cadmium (Cd), Chrome (Cr), Cobalt (Co), Lead (Pb), Antimony (Sb), and Nickel (Ni) data. [l]for Copper (Cu) data. m – for Mercury (Hg) data. [n]concentrations of nine heavy metals, above the standard guideline of safe limits. [o]negative statistically significant trends of NDVI, detected during 2000–2015 using the MK test and *Sen's slope* estimator. [p]negative statistically significant trends of GTD, detected during 2004–2013 using the MK test and *Sen's slope* estimator. For trend-based data of soil organic carbon loss, vegetation degradation and groundwater decline, the confidence level of the MK (two-tailed) test was set to a *p*-value threshold ≤ 0.1, which includes both highly statistically significant trends (for *p*-values ≤ 0.05) and trends with lower statistical significance (*p*-values between 0.05 and 0.1).

only by the ability of wind friction to exceed the entrainment threshold. Finally, for a given pixel we converted the one-dimensional $Q$ (g m$^{-1}$ s$^{-1}$) to an areal wind erosion (g m$^{-2}$ s$^{-1}$, subsequently converted to t ha$^{-1}$ yr$^{-1}$) by dividing by the length scale of the pixel. Details about the $Q$ modelling and the data layers used can be found in Chappell et al.[13]. For this study, we reprocessed the data of that original study so that we could export high-resolution (500 m) data for use in the subsequent analyses with other degradation processes (Table 3).

Soil nutrient imbalances were obtained based on geospatial data on nitrogen (N, in kg/ha) and phosphorus (P, in mg/kg) soil content and flows, which allowed to identify N and P surplus (that can indicate, for instance, the risk of pollution through overfertilization) and deficit (a likely decrease in land productivity) conditions (Table 3). For N, the first step involved the identification of an operative safe space as illustrated by Quemada et al.[33], beyond which environmental impacts may arise from both 1) excessive N surplus (that is N input – crop export),

leading to detrimental losses to air and water, and 2) very high N use efficiency (NUE, that is the ratio between crop export and N inputs) that can mine soil fertility and reduce the productivity. According to Quemada et al.[33]. and expert opinions, we defined an operative N safe space having NUE < 0.9 and <50 kg/ha, applying these thresholds to the dataset derived from DayCent continental simulations[34,35] extended at 1 km gridded level. Therefore, N imbalance was defined in this research as having surplus (>50 kg/ha) or NUE ( > 0.9) deficit (Table 3).

For the P data, we combined both the soil available P and budget (P input – export by crops) to define P imbalance conditions, since sorption/desorption processes vary widely in different soils, making the available P only an approximate indicator of P status (due to also legacy effects and its low reactivity in soil). Basically, we defined two P conditions that can threaten the agricultural land productivity, namely 1) a surplus level, when P available in soil is >50 mg/kg and there is a positive budget, and 2) a deficit condition, when P is <25 mg/kg and there is negative budget[36,37] (Table 3). After separately classifying N and P raster data, the four resulting classes were merged as a single raster with soil nutrient imbalances, which is itself can be considered an important facet of LD.

Soil pollution via heavy metals was processed using separate raster data of nine toxic substances (As, Cd, Cr, Co, Pb, Sb, Ni, Cu, and Hg, with values in mg/kg) that were spatially predicted in Europe[38–40]. The nine rasters, generated in several data sources based on Land Use/ Land Cover Area Frame Survey (LUCAS) topsoil database[41] and other auxiliary data[38–40], were first classified using some critical thresholds proposed for each harmful substance (Table 3). Subsequently, heavy metal soil pollution was processed as a single layer by joining/intersecting the individual rasters with delimited critical concentrations in topsoils. Selecting European areas with high concentrations of heavy metals, above the standard guideline thresholds of toxic elements[38,42] (Table 3), was essential for highlighting soil contamination, which is a threat to soil-based ecosystem services.

The vegetation degradation raster was obtained based on Normalized Difference Vegetation Index (NDVI) data, extracted and converted as annual values from the Moderate Resolution Imaging Spectroradiometer (MODIS) Terra MOD13A1 product[43]. This process was examined using the Mann-Kendall (MK) test[44,45] and *Sen's slope* estimator[46,47] in the analysis of annual NDVI trends, at pixel level (Table 2). Vegetation degradation was detected based on negative NDVI trends (which can define the areas with agricultural vegetation affected by devitalization, decreases in density/consistency or biomass decline), identified as statistically significant at the *p*-value ≤ 0.1 (a threshold that includes strong significant trends, for *p*-values < 0.05, but also the trends with lower statistical confidence, for *p*-values between 0.05 to 0.1)[48] (Table 3).

The groundwater decline modelling was based on groundwater table depth (GTD, in m) simulated using a global hydrology model[49], at ~1-km grid size and 1-hour time steps over 10 years (Table 2), driven by observed vegetation biomass (Leaf Area Index from MODIS) and European Centre for Medium Range Weather Forecast (ECMWF) ERA5 reanalysis atmosphere[50]. The model simulates soil water infiltration solving the 1D Richards' equation in columns discretized with 40 layers that get thicker with depth, down to 1 km deep. It includes groundwater dynamics, with the water table being the lower boundary condition of the soil columns. Vertically integrated groundwater lateral flows are calculated based on Darcy's law, driven by water table horizontal gradients. The model represents the 2-way exchange between groundwater and rivers or wetlands, river flow and inundation, and dynamic plant root uptake[49]. The model has been validated with flux tower observations of evapotranspiration and gage observations of stream flow. The model output was saved at monthly GTD values, but averaged here into annual values for detecting trends over a decade, using the MK test[44,45] and *Sen's slope* estimator[46,47] (Table 3). The two statistical procedures were applied for detecting negative GTD pixel

trends (which can suggest depleting or decreasing groundwater resources, with negative consequences for agricultural plant growth), statistically significant at the *p*-value ≤ 0.1 confidence level (Table 3).

The last raster layer, of aridity, was processed by computing an Aridity Index (AI, in mm/mm) (Table 3) based on mean multiannual precipitation (P, in mm) and potential evapotranspiration (PET, in mm) data, extracted from CHELSA (Climatologies at high resolution for the earth's land surface areas) database[51,52]. The Aridity Index, calculated as a ratio between the two climatic parameters (AI = P/PET), defines four types of drylands (lands affected by aridity) below the 0.65 mm/mm threshold – dry sub-humid (AI values between 0.65 and 0.5 mm/mm), semi-arid (0.5–0.2 mm/mm), arid (0.2–0.03 mm/mm) and hyper-arid (<0.03 mm/mm) lands[2,53]. In Europe, the aridity layer was produced based on dry sub-humid, semi-arid, and arid climate conditions, the presence of which on the continent can limit agricultural productivity through constant dryness or climate-induced desertification.

### Final data modelling

All 12 collected/processed spatial databases were finally processed at 500 m (an approximately intermediate spatial resolution in the variety of the original data resolution) (Table 3) and were structured/prepared into 2 general classes, named "Non-critical" and "Critical" (Fig. 1). The "Critical" class of each examined process was mapped using critical thresholds documented in literature, over/under which each LD process triggers the reduction/loss of agricultural land productivity (Table 3). In addition to the rigorous documentation from scientific literature, the critical thresholds were defined according to environmental criteria for healthy/unhealthy soil conditions (most notably for soil erosion, soil salinization, loss of soil organic carbon, and soil compaction), reported in the draft of the Soil Monitoring Law proposed by the European Commission (on July 5th, 2023) and currently under discussion in the European Parliament[54].

The "Critical" class, which highlights high/severe degradative conditions in agricultural landscapes (Table 3), was included in the final spatial data modelling, in order to obtain a relevant indicator for the continental assessment of multiple and convergent LD pathways. More specifically, by superimposing/intersecting the 12 datasets, we obtained the Land Multi-degradation Index (LMI), which indicates the simultaneous presence (co-occurrence) of degradative processes, at the pixel level. LMI (500 m pixel size) was processed strictly at the level of the agricultural boundaries in Europe, which were extracted from the CORINE Land Cover (CLC) database, 2018 edition (the most up-to-date)[55]. The CLC dataset used here includes four general agricultural classes, namely arable land (2.1.1, 2.1.2, and 2.1.3 codes in the CLC nomenclature), permanent crops (2.2.1, 2.2.2, 2.2.3), pastures (2.3.1) and heterogeneous agricultural areas (2.4.1, 2.4.2, 2.4.3, 2.4.4)[55].

LMI has been investigated at the level of all agricultural classes, but also strictly within arable lands, to emphasize the co-occurrence pattern of processes in the most important agricultural environments for European food security. Also, this index was processed in all 40 countries, despite the fact that for 3 processes (soil acidification, soil compaction, and soil pollution via heavy metals) there was no geospatial data for Norway, Switzerland, Balkan countries, Cyprus, and Malta (Fig. 1) (for these countries, in the Supplementary Information section, LMI was obtained based on the 9 remaining processes with available data).

Finally, LMI was investigated and interpreted based on five general classes, which emphasized very low (a single process present at pixel level), low (co-occurrence of 2 LD processes), medium (3), high (4) and very high (≥5) degradative conditions in agricultural/arable lands (Fig. 2). We set these classes considering the LMI histogram (in which a data distribution of >90% condensed in the interval 1 to 5 synergistic processes was observed) (Fig. 2b,e), the natural breaks classification method[56] – which revealed roughly similar classes, but also the LMI value arithmetic mean, which showed just over 2

co-occurring processes in Europe (close to an intermediate value in the 1–5 interval used to set LMI classes). In addition to the 3 statistical criteria considered simultaneously, we set the 5 classes also with the aim to obtain an easy/fast interpretation of the LMI. All data processing and graphic analyses of this study were performed using various software, like R-package[57], ArcGIS[58], or Inkscape[59].

## Data quality and limitations

The quality of this study's data can be evaluated via three key aspects. Firstly, the reliability of the 6 datasets directly acquired is supported by the published studies referenced in this paper (data source indicated for processes no. 1, 3, 4, 5, 6, and 8, in Table 3), where uncertainties and errors were generally addressed. Secondly, we considered the quality of the data for the other 6 processes modelled here, by using reliable pre-existent data (input data of processes no. 2, 7, 9, 10, 11, and 12, which were also checked in the sources featured in Table 3, in terms of uncertainties and errors) in obtaining the final geospatial layers. Thirdly, for the 2 layers created in this study (no. 10 and 11 in Table 3), we exclusively selected statistically significant trends, which limits uncertainties/errors in these particular cases.

Some potential limitations may exist in our methodological approach. These may emerge from the different spatial/temporal resolution (1) and metrics (2) of the geospatial data, from the choice in thresholds that define "Critical" classes of the twelve processes (3), from assigning equal contributions to all layers in computing LMI (4), or from integrating only 75–83% of all input layers for LMI computation (Supplementary Information), in the case of some European countries (5). In the first case (1), it would have been ideal to use datasets with similar/identical spatial and temporal resolutions, which was however impossible, given the high number of data layers used, with different technical (pixel size and time periods) characteristics available in the literature (Table 3).

In the second instance (2), it would have been better if soil compaction and soil pollution via pesticides had a quantitative nature (similarly to all the other 10 databases), instead of a qualitative (adimensional) measurement unit (Table 3), but in the case of the two processes we were constrained by the unavailability of other (quantitative) data in literature. Also, another (apparent) shortcoming on this second point could be linked to the geospatial layers that were obtained based on some multi-temporal data, but which were interpreted based on different metrics. More precisely, vegetation degradation and groundwater decline were assessed as LD pathways based on negative annual trends (as per the reasoning presented in the section dedicated to data acquisition/preparation), while the aridity layer was examined through a mean multiannual ratio (of climate data) (Table 3). However, we argue this choice is more appropriate, as the mere presence of aridity conditions (and not necessarily the intensification of this process, which could have been detected based on geostatistical trends) is sufficient to trigger some major drivers of degradation, like desertification[3].

In the third case (3), the critical thresholds of some factors may be debated – e.g. severe erosion rates[60] are considered as those exceeding the threshold of 10 t ha$^{-1}$ yr$^{-1}$, while in other studies[8,61], the unsustainable level of this process (examined in the long term in relation to soil formation rates) is adjusted to >2 t ha$^{-1}$ yr$^{-1}$, the threshold selected in this research (Table 3). Since our applied methodology required choosing a unique critical threshold for each layer of degradation, we eventually set "Critical" classes by consulting a high number of relevant papers and scientific reports (only a few of which were exemplified in this study), which supported the thresholds selected for this approach (Table 3).

In the fourth case (4), we attributed equal importance to all drivers of degradation, although in reality, some processes (e.g. soil salinization) could trigger more severe effects in the decrease of agricultural productivity, compared to other degradative factors (e.g. vegetation degradation). However, we avoided a weighting of the factors in modelling LMI, considering that, in principle, each process can lead to multiple negative effects for land productivity (Table 2), especially since it is almost impossible to determine the exact impact of each process in the land multi-degradation mechanism.

Finally (5), due to the unavailability of geospatial data, we omitted up to 25% (three LD processes – soil acidification, soil compaction, soil pollution via heavy metals) of all input layers used to model LMI results, which were presented in detail (for all 40 countries) in the Supplementary Information section. Nevertheless, these missing data affected (potentially underestimated) LMI results in the case of a limited number of European countries (Supplementary Figs. 2–4 and Supplementary Tables 1–13), i.e. the Balkan states, Switzerland, Norway (with all three processes not integrated in the LMI, but still with three-quarters of all databases available to obtain the final results of land multi-degradation) and, partially, Cyprus and Malta (only two processes missing – soil acidification and soil pollution via heavy metals) (Fig. 1).

## Data uncertainty and sensitivity

To assess the effect of varying thresholds on the binary classification adopted to obtain the LMI map (which was a crucial methodological phase of our approach), an uncertainty and sensitivity analysis was performed. More specifically, through an operation of layer stacking, the multiple data used to derive the twelve LD binary layers (maps) were combined in a single matrix of original inputs. For each of one the input maps, random simulations ($n = 20,000$) were adopted by varying the threshold initially set and using values drawn from a normal distribution, with the numerical value of the threshold as the mean and a value of one-tenth of the threshold as standard deviation. The matrix was then reclassified into binary classes according to the thresholds randomly defined. Subsequently, the row values were summed up to obtain the LMI values under different thresholds. Given the reclassification step, the LMI still varied from 0 to 10, but in some of the rows, its values (under the same initial conditions) might have varied due to the different classification.

Finally, the LMI values derived were used as a dependent variable in a Random Forest (RF) classification model (1000 decision trees and 20 repetitions)[62] that was applied in a R statistics enviroment[63], using the initial input maps as covariates. The RF model was used for its ability to produce classification probabilities (so predicting with which probability a pixel falls within a given class), thus providing an estimate of how much varying threshold results in a different final classification (Supplementary Fig. 1). Moreover, the RF results obtained (Supplementary Fig. 1) can be leveraged to assess how much each of the initial variables influences the classification, consequently offering a valuable sensitivity analysis for our LMI findings.

## Reporting summary

Further information on research design is available in the Nature Portfolio Reporting Summary linked to this article.

# Data availability

The data supporting the findings of this research are available in the article and its Supplementary Information file. Also, the source data for the graphs of the figures are provided as a Source Data file. At the same time, the raster data (GeoTIFF format) of land degradation processes and land multi-degradation in Europe will be freely available through the European Soil Data Centre (ESDAC), the institutional soil data platform of the European Commission's Joint Research Centre (https://esdac.jrc.ec.europa.eu/). Additional data can be provided by the corresponding author upon request. Source data are provided with this paper.

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

## Acknowledgements
This work was funded by the EU's NextGenerationEU instrument through the National Recovery and Resilience Plan of Romania – Pillar III-C9-2022-I8, managed by the Ministry of Research, Innovation and Digitalization, within the project entitled "Complex modelling of multiple land degradation processes in Europe. Towards an integrative scientific framework for sustainable land management across the continent", contract no. 760051/23.05.2023, code CF 216/29.11.2022. This project provided funding for R.P., P.B., M.N., B.R., C.P., and G.B. Also, P.B. has received funding from the Swiss State Secretariat for Education, Research and Innovation (SERI), grant agreement no. 101086179, AI4SoilHealth.

## Author contributions
R.P. conceptualized the study, designed and led the research, analyzed the study's results, and wrote the paper. P.B. provided continental data, performed the computational analysis of some geospatial datasets, and contributed to the writing of the paper. P.P., E.L., A.C., G.M.M., F.M. and J.P. provided multiple geospatial datasets at the continental level, made contributions to the study methodology, and gave feedback on data interpretation. C.B. performed the continental analysis of data uncertainty and sensitivity. M.N. produced the figures, i.e. designed the graphs and mapped the various land degradation processes and land multi-degradation results obtained at European scale. B.R., C.P., M.D., G.B., I.A.N. and M.V.B. processed the raw geospatial data, modelled the final results, and performed the statistical analyses of the study.

## Competing interests
The authors declare no competing interests.
