## [Peer Review File · Nature Communications]

REVIEWER COMMENTS

Reviewer #1 (Remarks to the Author):

The study addresses the important topic of land degradation assessment. The authors have used an impressive amount of spatial layers to represent the complexity of land degradation processes. This is highly appreciated and certainly needed to improve land degradation neutrality accounting.

I find the systematic mapping of convergent LD processes of high relevance for policy making as it enables to diagnose the degradation pathways over regional and local hotspots. The overall framework and design of the Land Multi-degradation Index is relatively simple and for good reason, as it is meant to serve environmental accounting. However this should not prevent the careful assessment of data and methodological choices upon which the LMI is built. Here the authors rely entirely on thresholds applied to 12 LD indicators, thresholds that are based on literature for 10 out of 12. This raises several concerns with regards to:

(1) the over-simplified picture of land degradation when relying on multiple binary indicators / thresholds (so-called critical / non-critical values). I understand the need for simplifying a complex process for environmental accounting, however I would have welcomed a third category corresponding to mild degradation and/or a throughout sensitivity analysis demonstrating a low dependency of LMI to the final selection of LD thresholds ;

(2) the selection and documentation of threshold values. It is unclear how some final threshold values were selected e.g., for soil water erosion the authors refer to a study on tolerable soil erosion that indicates another threshold than the one used here (1.4 t ha⁻¹ yr⁻¹ instead of 2 t ha⁻¹ yr⁻¹). Again knowing the potentially large dependency of the LMI on the selected indicators and applied threshold values, I would recommend the authors to provide further justification in relation to the thresholds they have used and their applicability at continental scale ;

(3) Uncertainty estimation. Linked with the previous points, the authors did not provide uncertainty estimates for the LMI map, which is unfortunate considering the fact that the product has the potential/ambition to be used for area estimates and accounting at country level. Performing a sensitivity analysis on threshold values to estimate its impact on the LMI (and derived country-level statistics) would have been highly beneficial to understand the robustness of the final LMI map and associated products (e.g., figure 3).

Furthermore the computation of LMI over countries (or regions) where data are missing should be avoided as it has a high risk of under-estimating land degradation (e.g., 1/4 of data missing for several Balkan countries). These countries should simply be excluded from the final mapping, or the LMI scale adjusted accordingly.

Reviewer #3 (Remarks to the Author):

The comprehensive evaluation of LD is useful for protecting ecological condition, ensuring food security, maintaining ecosystem services, and preserving biodiversity. Research on the LD definition, LD processes and its drivers is essential for preventing LD and restoring degraded land.

This manuscript has collected (or produced) data on 12 LD processes in Europe, covering major agricultural ecosystem LD issues. Additionally, the ms created LMI to evaluate the degree of land multi-degradation for European agricultural land. The evaluation of multi-dimensional LD could enhance the knowledge of LD processes in European agricultural land, and provide rich LD data sources in European agricultural land.

The research design is reasonable and clear, except three major issues as below:

1. The research on definition, drivers and evaluation of degree of LD pose major challenges for land science. This MS introduces the data accuracy for each LD process separately. However, evaluating the degree of LD based on all LD processes using the LMI could potentially introduce further uncertainties and errors. Could such cumulative uncertainties of the multi-LD process potentially affect the scientific application for decision-makers? How was the final multi-LD process evaluated for this region?
2. This ms assign equal importance to each individual LD process in LMI mainly because it is not easy to determine the exact multi effect of each process in land multi-degradation mechanism. Actually, I find this difficult to understand because evaluating and measuring the effects of LD is challenging, so the equal importance is better than other methods? Further discussion on the applicability of LMI in other regions or research, as well as the limitations of this method, may be necessary.
3. This MS has introduced a clear spatial pattern of multi-LD process. However, what factors contribute to such a pattern in Europe? For example, why large parts of the agricultural areas are exposed to one, two and three drivers of LD (page 4, the second paragraph from bottom). Additionally, an overview of the drivers of interacting LD process across the continent could provide decision-makers with more specific information. An introduction to the drivers of LD in the study region would be beneficial.

Some minor modifications require further checking.

1. Why did author evaluate multi-LD processes only for agricultural land?
2. Page 2, second paragraph, should the full name be indicated when an institutional acronym is mentioned for the first time? Like IPCC.
3. In Table 3, the time period varies for each LD process. For example, some indicators are available for only one year (e.g., water erosion, 2023), while others have dynamic datasets (wind erosion, 2001-2021). How did the authors standardize the data time? Did you calculate the mean values for wind erosion by using 2001-2021 data?

A unifying modelling of multiple land degradation pathways in Europe

- Response to reviewer comments -

REVIEWER #1 (Remarks to the Author):

Comment 1 (general remarks)

The study addresses the important topic of land degradation assessment. The authors have used an impressive amount of spatial layers to represent the complexity of land degradation processes. This is highly appreciated and certainly needed to improve land degradation neutrality accounting.

I find the systematic mapping of convergent LD processes of high relevance for policy making as it enables to diagnose the degradation pathways over regional and local hotspots. The overall framework and design of the Land Multi-degradation Index is relatively simple and for good reason, as it is meant to serve environmental accounting. However this should not prevent the careful assessment of data and methodological choices upon which the LMI is built. Here the authors rely entirely on thresholds applied to 12 LD indicators, thresholds that are based on literature for 10 out of 12. This raises several concerns with regards to:

Response 1

Thank you for your overall appreciation of our research!

Please see the responses below, where we tried to carefully address your valuable observations.

Comment 2

(1) the over-simplified picture of land degradation when relying on multiple binary indicators / thresholds (so-called critical / non-critical values). I understand the need for simplifying a complex process for environmental accounting, however I would have welcomed a third category corresponding to mild degradation and/or a throughout sensitivity analysis demonstrating a low dependency of LMI to the final selection of LD thresholds;

Response 2

We understand your concern, given that, in theory, an intermediate/mild class could indeed be useful in susceptibility/sensitivity analyses of land degradation. However, we did not introduce a third category in our modelling, for practical (1) and methodological (2) reasons. In the first case (1), we opted for an approach limited to 2 classes (binary classification) to align our results to the EU Soil Monitoring Law draft (https://environment.ec.europa.eu/publications/proposal-directive-soil-monitoring-and-resilience_en), currently under discussion in the European Parliament. In the EU Soil Monitoring Law draft, specific thresholds are also provided for the identified soil threats. When possible, as further discussed in the next point (comment), our thresholds are aligned with the ones of the EU Soil Monitoring Law. For more details, please see the answer provided to the next comment.

In the second case (2), our methodology is different from other approaches based on distinct (susceptibility) models, which allow the use of three classes of degradation or even more. More exactly, some land degradation susceptibility models, like DISMED or MEDALUS, are based on the classification of input geospatial data into several sensitivity classes (e.g. low, moderate and high land sensitivity to degradation / desertification), according to the information that can be found in international literature or in some papers

published by the first author of this study (see, for example, “Prăvălie, et al., 2017. Quantification of land degradation sensitivity areas in Southern and Central Southeastern Europe. New results based on improving DISMED methodology with new climate data. *Catena* 158, 309–320, <https://doi.org/10.1016/j.catena.2017.07.006>” or “Prăvălie, et al., 2020. Spatial assessment of land sensitivity to degradation across Romania. A quantitative approach based on the modified MEDALUS methodology. *Catena* 187, <https://doi.org/10.1016/j.catena.2019.104407>”).

However, the major methodological shortcoming of susceptibility models is that the input data does not represent actual land degradation processes, but various ecological and socio-economic conditions whose synergistic effects can potentially lead to land degradation. Essentially, these exemplified models do not concretely (directly) detect land degradation, but rather the fact this environmental issue may exist, according to the brief specifications featured in the introduction (“*While other studies explored the complex nature of LD based on certain multicriteria susceptibility models applied at global²⁵ or European²⁶ scales, they too generated results that did not focus on actual LD processes, but on examining the synergic effects of various ecological and socio-economic conditions potentially leading to LD.*”).

Therefore, we consider that the methodological approach used in this study is superior to sensitivity models, since our approach includes real land degradation processes. However, in this context, we had to apply a binary classification of input geospatial data, based on setting strictly 2 classes for each layer (process) of land degradation, which emphasize the presence or absence of the land processes’ severity. This methodological philosophy, although simple, was the best way for a clear detection of concurrent processes, via LMI. Thus, considering the methodological scheme of LMI, focused on quantifying the number of convergent processes, we hope you will see that the third class cannot be introduced in our modelling.

Comment 3

(2) the selection and documentation of threshold values. It is unclear how some final threshold values were selected e.g., for soil water erosion the authors refer to a study on tolerable soil erosion that indicates another threshold than the one used here (1.4 t ha⁻¹ yr⁻¹ instead of 2 t ha⁻¹ yr⁻¹). Again knowing the potentially large dependency of the LMI on the selected indicators and applied threshold values, I would recommend the authors to provide further justification in relation to the thresholds they have used and their applicability at continental scale;

Response 3

Kindly note that a brief justification of the land degradation critical thresholds is already provided at the Table 3 of the article:

“b – critical thresholds over/under which each land degradation process triggers the reduction or loss of agricultural land productivity; these thresholds used for modelling agricultural land multi-degradation were documented and set for each process based on scientific literature: water and wind erosion^{10,80,81}, soil organic carbon loss^{2,6}, soil salinization⁵⁸, soil acidification^{4,38}, soil compaction⁶⁰, soil nutrient imbalances^{61,64,65,79}, soil pollution via pesticides²⁹, soil pollution via heavy metals^{50,68} and aridity^{2,57}; as no concrete thresholds were found in literature for vegetation degradation and groundwater decline, in these two cases some critical thresholds/classes were set in accordance with the reasoning explained in o and p”.

We believe that the information reported in the caption of Table 3 should be adequate to justify the thresholds used, which were adopted after carefully checking international literature. And please note that,

for each threshold used, we only cited a few relevant sources (in the caption of Table 3), but in reality there are numerous scientific papers that confirm the same critical thresholds of land degradation processes.

However, your useful comment gave us the opportunity to add a short phrase in this revised paper (in “*Final data modelling*”), to better highlight the adopted thresholds in our modelling:

“In addition to the rigorous documentation from scientific literature, the critical thresholds were defined according to environmental criteria for healthy / unhealthy soil condition (most notably for soil salinization, soil erosion, loss of soil organic carbon, and soil compaction), reported in the draft of the Soil Monitoring Law proposed by the European Commission (on July 5th, 2023) and currently under discussion in the European Parliament⁸²”.

Thus, besides the in-depth documentation of specialized literature, some of the adopted thresholds are aligned to the criteria reported in the draft of the *Soil Monitoring Law*, proposed by the European Commission on July 5th, 2023 (EC, 2023) and currently under discussion in the European Parliament. In fact, when applicable (most notably for salinization, soil erosion, loss of soil organic carbon, and subsoil compaction), the thresholds reported in our study are aligned to the criteria for healthy soil condition identified at EU level by a team of experts established within the EU Soil Mission ‘*A Soil Deal for Europe*’ framework (experts from the European Environmental Agency, Directorate-General for Agriculture and Rural Development and members of the European Soil Partnership).

Concerning the mentioned soil erosion threshold, it is consistent with the value of 2 t ha⁻¹ yr⁻¹ proposed in the *EU Soil Monitoring Law* (ANNEX I, see below). The threshold was defined according to the soil formation values reported in scientific literature, among which there is the study of Verheijen et al. 2009 (“Tolerable versus actual soil erosion rates in Europe”), but also other studies reporting higher soil formation rates (e.g. “Egli et al., 2014. Soil formation rates on silicate parent material in alpine environments: Different approaches-different results? *Geoderma*, 213, 320–333”). To our knowledge, the Directorate-General for Agriculture and Rural Development of the European Commission defined the threshold to 2 t ha⁻¹ yr⁻¹ for the Soil Monitoring Law, and we have applied it accordingly.

ANNEX I

SOIL DESCRIPTORS, CRITERIA FOR HEALTHY SOIL CONDITION, AND LAND TAKE AND SOIL SEALING INDICATORS

For the purposes of this Annex, the following definitions shall apply

- (1) ‘reverse land take’ means the conversion of artificial land into natural or semi-natural land;
- (2) ‘net land take’ means the result of land take minus reverse land take.

Aspect of soil degradation	Soil descriptor	Criteria for healthy soil condition	Land areas that shall be excluded from achieving the related criterion
Part A: soil descriptors with criteria for healthy soil condition established at Union level			
Soil erosion	Soil erosion rate (tonnes per hectare per year)	$\leq 2 \text{ t ha}^{-1} \text{ y}^{-1}$	Badlands and other unmanaged natural land areas, except if they represent a significant disaster risk

The ANNEX I ‘SOIL DESCRIPTORS, CRITERIA FOR HEALTHY SOIL CONDITION, AND LAND TAKE AND SOIL SEALING INDICATORS’ of the Soil Monitoring Law (https://environment.ec.europa.eu/system/files/2023-07/ANNEXES%20to%20the%20proposal%20for%20a%20Directive%20of%20the%20European%20Parliament%20and%20of%20the%20Council_COM_2023_416_final.pdf).

Comment 4

(3) Uncertainty estimation. Linked with the previous points, the authors did not provide uncertainty estimates for the LMI map, which is unfortunate considering the fact that the product has the potential/ambition to be used for area estimates and accounting at country level. Performing a sensitivity analysis on threshold values to estimate its impact on the LMI (and derived country-level statistics) would have been highly beneficial to understand the robustness of the final LMI map and associated products (e.g., figure 3).

Response 4

This is a fundamental aspect, indeed, which is why we worked hard to address it in this revision. We used the outcomes of the Random Forest model to estimate the uncertainty associated with LMI spatial results. We explained the uncertainty estimation methodology in a new section added to this revised paper (in the “Methods” section), titled “Data uncertainty and sensitivity” (we also mapped the uncertainty results in a newly added figure – Supplementary Fig. 1):

“To assess the effect of varying thresholds on the binary classification adopted to obtain the LMI map (which was a crucial methodological phase of our approach), an uncertainty and sensitivity analysis was performed. More specifically, through an operation of layer stacking, the multiple data used to derive the twelve LD binary layers (maps) were combined in a single matrix of original inputs. For each of one the input maps, random simulations ($n = 20,000$) were adopted by varying the threshold initially set and using values drawn from a normal distribution, with the numerical value of the threshold as the mean and a value of one tenth of the threshold as standard deviation. The matrix was then reclassified into binary classes according to the thresholds randomly defined. Subsequently, the row values were summed up to obtain the LMI values under different thresholds. Given the reclassification step, the LMI still varied from 0 to 10, but in some of the rows its values (under the same initial conditions) might have varied due to the different classification.

Finally, the LMI values derived were used as a dependent variable in a Random Forest (RF) classification model (1,000 decision trees and 20 repetitions)⁸⁹ that was applied in a R statistics environment⁹⁰, using the initial input maps as covariates. The RF model was used for its ability to produce classification probabilities (so predicting with which probability a pixel falls within a given class), thus providing an estimate of how much a varying threshold results in a different final classification (Supplementary Fig. 1). Moreover, the RF results obtained (Supplementary Fig. 1) can be leveraged to assess how much each of the initial variables influences the classification, consequently offering a valuable sensitivity analysis for our LMI findings.”.

“**Supplementary Figure 1.** Uncertainty and sensitivity of LMI results in Europe. **a–e**, Spatial distribution of uncertainty presented as probability (%) for LMI classes 1–5. **f**, Covariates used in applying the RF model, in order to predict uncertainties of LMI classes and provide a sensitivity analysis. Notes: probability (a–e) ranges from 0 (low) to 100 (high), and was obtained by applying a RF classification model (1,000 decision trees and 20 repetitions); covariates were used in the RF classification model for predicting with which probability a pixel falls within a given LMI class; covariate importance is expressed by the Mean Decrease Gini, which is a measure of how each input variable contributes in the prediction of LMI classes; the higher the value of mean decrease Gini coefficient, the higher the importance of the variable in the model is, thus reflecting the sensitivity of each variable for predicting LMI classes; RF means Random Forest, while LMI is the acronym for Land Multi-degradation Index.”

Also, we briefly presented the spatial results of the Random Forest model in the section “*Land multi-degradation pattern in Europe*”:

“*In order to statistically consolidate all these LMI results, we explored the potential uncertainties of our modelling, which may primarily result from defining the critical thresholds of multiple LD processes (see*

Methods). Thus, the possible uncertainties associated to the threshold-driven LMI values were quantified and mapped across Europe, using a Random Forest classification model (see Methods).

The results emphasized an overall prediction error of 17.6% (out-of-bag error) throughout the continent. Essentially, the uncertainties were defined as the probability that each modelled pixel falls in one of the LMI classes, according to Supplementary Fig. 1. The findings on the uncertainty (and sensitivity) analysis allowed us to define the geographical variability (Supplementary Fig. 1) of the LMI classes (associated to thresholds different from those that were defined according to scientific and policy related literature, as detailed in Methods), but also to estimate the prediction error associated to each LMI class area resulted in this study (Table 1).

Acknowledging some degree of uncertainty (Table 1, Supplementary Fig. 1), our multi-degradation approach becomes a better tool that can be highly useful for various EU policies (see Policy implications). Scientifically, the uncertainty modelling framework helps moving closer towards robust, repeatable, and open data science to communicate with adjacent disciplines and better deal with complex LD challenges.”.

We also adapted Table 1, by adding error ranges of LMI class areas, which resulted from the application of the Random Forest model:

“**Table 1.** Spatial extent (in km² and %) of LMI classes in agricultural/arable environments of Europe

No.	LMI classes (number of co-occurring processes)	Agricultural lands		Arable lands	
		km ²	%	km ²	%
1	No degradation (0) ^a	120,963 (± 29,031)	6.16 (± 1.48)	66,410 (± 15,938)	6.07 (± 1.46)
2	Very low degradation (1)	523,824 (± 167,624)	26.70 (± 8.54)	287,879 (± 92,121)	26.33 (± 8.43)
3	Low degradation (2)	678,224 (± 149,485)	34.57 (± 7.62)	372,937 (± 82,198)	34.11 (± 7.52)
4	Medium degradation (3)	440,473 (± 66,655)	22.45 (± 3.40)	244,081 (± 36,936)	22.32 (± 3.38)
5	High degradation (4)	155,265 (± 25,020)	7.91 (± 1.28)	94,654 (± 15,253)	8.66 (± 1.39)
6	Very high degradation (≥ 5) ^b	43,352 (± 8,919)	2.21 (± 0.45)	27,460 (± 5,650)	2.51 (± 0.52)

Notes: a – agricultural/arable lands unaffected by degradation processes; b – most frequently five concurrent processes, according to the LMI histograms for agricultural/arable areas (Fig. 2b,e); % – the percentage-based area of the number of convergent processes (0, 1, 2, 3, 4, ≥5), related to the absolute area of continental agricultural (1,962,101 km²)/arable (1,093,421 km²) lands; the values in parentheses (±) are error ranges obtained by applying a Random Forest classification model (see Methods); all these European statistics were extracted after excluding the countries with incomplete data for LMI modelling (Fig. 2).”

Therefore, we consider that all these uncertainty investigations strengthen our modelling and considerably increase the statistical quality of our results.

Comment 5

Furthermore, the computation of LMI over countries (or regions) where data are missing should be avoided as it has a high risk of under-estimating land degradation (e.g., 1/4 of data missing for several Balkan countries). These countries should simply be excluded from the final mapping, or the LMI scale adjusted accordingly.

Response 5

Indeed, two-three input variables (soil acidification, soil compaction, soil pollution via heavy metals) out of twelve were missing for LMI modelling in some European countries. We cannot adjust the LMI scale in this case, but we agree with your valuable observation and that is why we tried to address it accordingly. Therefore, following your comment, we masked those countries from the main body of the article (see Figures 2, 3 and 4) and adjusted the continental statistics in Table 1, after excluding countries with some

missing data. The new statistics were highlighted in the text, in the section “*Land multi-degradation pattern in Europe*”.

However, considering that for those countries (generally Balkan states) with some missing data we still used a relatively large number of land degradation processes (75–83% or 9–10 input layers out of 12) in calculating the LMI, we kept the original data in the Supplementary Information (SI) section. Thus, we moved the previous figures (2, 3 and 4) and Table 1 to SI, and we kept all the other initial tables that include the data extracted at the level of the 40 European countries. We chose this option because we are confident that keeping the LMI results also for those particular countries still provides a good / very good picture of land degradation, since a large number of input data were available for LMI modelling. We clearly highlighted this argument in the explanatory notes of Figures 2–4 transferred to SI:

“compared to Fig. 2, the LMI values are spatialized here for all analysed states, considering that in some cases with incomplete data (Norway, Switzerland, Balkan countries, Cyprus and Malta) we still used a large number of process databases (9–10 input layers out of 12) in computing LMI (see Methods); consequently, LMI still has the potential to provide a good/very good picture of land degradation in these particular countries, even if their LMI values should be interpreted with some caution” (similar notes also for Supplementary Figures 3 and 4 / Supplementary Tables).

Nevertheless, we drew attention to the limitation of some missing data, even though the original data was transferred to SI. More exactly, in SI, in the informative notes of all tables (and figures), we clearly stated that results should be interpreted with caution for those particular countries with some missing data: “*LMI statistics should be interpreted with some caution within several European countries (Norway, Switzerland, Balkan countries, Cyprus and Malta), where only 9–10 of the 12 input geospatial databases (Fig. 1) were available for LMI computation*”.

In addition, we clearly highlighted this shortcoming once again in the article section titled “*Data quality and limitations*”: “*Some potential limitations may exist in our methodological approach. These may emerge from ..., or from integrating only 75–83% of all input layers for LMI computation (Supplementary Information), in the case of some European countries (5)*” and “*Finally (5), due to the unavailability of geospatial data, we omitted up to 25% (three LD processes – soil acidification, soil compaction, soil pollution via heavy metals) of all input layers used to model LMI results, which were presented in detail (for all 40 countries) in the Supplementary section. Nevertheless, these missing data affected (potentially underestimated) LMI results in the case of a limited number of European countries (Supplementary Figs. 2–4 and Supplementary Tables 1–13), i.e. the Balkan states, Switzerland, Norway (with all three processes not integrated in the LMI, but still with three quarters of all databases available to obtain the final results of land multi-degradation) and, partially, Cyprus and Malta (only two processes missing – soil acidification and soil pollution via heavy metals) (Fig. 1).*”.

REVIEWER #3 (Remarks to the Author):

Comment 1 (general remarks)

The comprehensive evaluation of LD is useful for protecting ecological condition, ensuring food security, maintaining ecosystem services, and preserving biodiversity. Research on the LD definition, LD processes and its drivers is essential for preventing LD and restoring degraded land.

This manuscript has collected (or produced) data on 12 LD processes in Europe, covering major agricultural ecosystem LD issues. Additionally, the MS created LMI to evaluate the degree of land multi-degradation for European agricultural land. The evaluation of multi-dimensional LD could enhance the knowledge of LD processes in European agricultural land, and provide rich LD data sources in European agricultural land.

The research design is reasonable and clear, except three major issues as below:

Response 1

Thank you for your overall positive feedback on our study!

Please see responses below, where we tried to adequately address your helpful comments.

Comment 2

1. The research on definition, drivers and evaluation of degree of LD pose major challenges for land science. This MS introduces the data accuracy for each LD process separately. However, evaluating the degree of LD based on all LD processes using the LMI could potentially introduce further uncertainties and errors. Could such cumulative uncertainties of the multi-LD process potentially affect the scientific application for decision-makers? How was the final multi-LD process evaluated for this region?

Response 2

We fully agree with you that such an uncertainty analysis is crucial in our work, which is why we did our best to follow your suggestion. More exactly, we applied the Random Forest model to estimate the uncertainty associated with LMI spatial results, according to the methodological explanations provided in a new section added to this revised paper (in the “Methods” section), titled “Data uncertainty and sensitivity” (we also mapped the uncertainty results in a newly added figure – Supplementary Fig. 1):

“To assess the effect of varying thresholds on the binary classification adopted to obtain the LMI map (which was a crucial methodological phase of our approach), an uncertainty and sensitivity analysis was performed. More specifically, through an operation of layer stacking, the multiple data used to derive the twelve LD binary layers (maps) were combined in a single matrix of original inputs. For each of one the input maps, random simulations ($n = 20,000$) were adopted by varying the threshold initially set and using values drawn from a normal distribution, with the numerical value of the threshold as the mean and a value of one tenth of the threshold as standard deviation. The matrix was then reclassified into binary classes according to the thresholds randomly defined. Subsequently, the row values were summed up to obtain the LMI values under different thresholds. Given the reclassification step, the LMI still varied from 0 to 10, but in some of the rows its values (under the same initial conditions) might have varied due to the different classification.

Finally, the LMI values derived were used as a dependent variable in a Random Forest (RF) classification model (1,000 decision trees and 20 repetitions)⁸⁹ that was applied in a R statistics environment⁹⁰, using the initial input maps as covariates. The RF model was used for its ability to produce classification probabilities (so predicting with which probability a pixel falls within a given class), thus providing an estimate of how much a varying threshold results in a different final classification (Supplementary Fig. 1). Moreover, the RF results obtained (Supplementary Fig. 1) can be leveraged to assess how much each of the initial variables influences the classification, consequently offering a valuable sensitivity analysis for our LMI findings.”.

“**Supplementary Figure 1.** Uncertainty and sensitivity of LMI results in Europe. **a–e**, Spatial distribution of uncertainty presented as probability (%) for LMI classes 1–5. **f**, Covariates used in applying the RF model, in order to predict uncertainties of LMI classes and provide a sensitivity analysis. Notes: probability (a–e) ranges from 0 (low) to 100 (high), and was obtained by applying a RF classification model (1,000 decision trees and 20 repetitions); covariates were used in the RF classification model for predicting with which probability a pixel falls within a given LMI class; covariate importance is expressed by the Mean Decrease Gini, which is a measure of how each input variable contributes in the prediction of LMI classes; the higher the value of mean decrease Gini coefficient, the higher the importance of the variable in the model is, thus reflecting the sensitivity of each variable for predicting LMI classes; RF means Random Forest, while LMI is the acronym for Land Multi-degradation Index.”

Also, we briefly presented the spatial results of the Random Forest model in the section “*Land multi-degradation pattern in Europe*”:

“*In order to statistically consolidate all these LMI results, we explored the potential uncertainties of our modelling, which may primarily result from defining the critical thresholds of multiple LD processes (see*

Methods). Thus, the possible uncertainties associated to the threshold-driven LMI values were quantified and mapped across Europe, using a Random Forest classification model (see Methods).

The results emphasized an overall prediction error of 17.6% (out-of-bag error) throughout the continent. Essentially, the uncertainties were defined as the probability that each modelled pixel falls in one of the LMI classes, according to Supplementary Fig. 1. The findings on the uncertainty (and sensitivity) analysis allowed us to define the geographical variability (Supplementary Fig. 1) of the LMI classes (associated to thresholds different from those that were defined according to scientific and policy related literature, as detailed in Methods), but also to estimate the prediction error associated to each LMI class area resulted in this study (Table 1).

Acknowledging some degree of uncertainty (Table 1, Supplementary Fig. 1), our multi-degradation approach becomes a better tool that can be highly useful for various EU policies (see Policy implications). Scientifically, the uncertainty modelling framework helps moving closer towards robust, repeatable, and open data science to communicate with adjacent disciplines and better deal with complex LD challenges.”.

We also adapted Table 1, by adding error ranges of LMI class areas, which resulted from the application of the Random Forest model.

“Table 1. Spatial extent (in km² and %) of LMI classes in agricultural/arable environments of Europe

No.	LMI classes (number of co-occurring processes)	Agricultural lands		Arable lands	
		km ²	%	km ²	%
1	No degradation (0) ^a	120,963 (± 29,031)	6.16 (± 1.48)	66,410 (± 15,938)	6.07 (± 1.46)
2	Very low degradation (1)	523,824 (± 167,624)	26.70 (± 8.54)	287,879 (± 92,121)	26.33 (± 8.43)
3	Low degradation (2)	678,224 (± 149,485)	34.57 (± 7.62)	372,937 (± 82,198)	34.11 (± 7.52)
4	Medium degradation (3)	440,473 (± 66,655)	22.45 (± 3.40)	244,081 (± 36,936)	22.32 (± 3.38)
5	High degradation (4)	155,265 (± 25,020)	7.91 (± 1.28)	94,654 (± 15,253)	8.66 (± 1.39)
6	Very high degradation (≥ 5) ^b	43,352 (± 8,919)	2.21 (± 0.45)	27,460 (± 5,650)	2.51 (± 0.52)

Notes: a – agricultural/arable lands unaffected by degradation processes; b – most frequently five concurrent processes, according to the LMI histograms for agricultural/arable areas (Fig. 2b,e); % – the percentage-based area of the number of convergent processes (0, 1, 2, 3, 4, ≥5), related to the absolute area of continental agricultural (1,962,101 km²)/arable (1,093,421 km²) lands; the values in parentheses (±) are error ranges obtained by applying a Random Forest classification model (see Methods); all these European statistics were extracted after excluding the countries with incomplete data for LMI modelling (Fig. 2).”

Therefore, we are confident that all these uncertainty investigations strengthen our modelling and considerably increase the statistical quality of our results.

Comment 3

2. This MS assign equal importance to each individual LD process in LMI mainly because it is not easy to determine the exact multi effect of each process in land multi-degradation mechanism. Actually, I find this difficult to understand because evaluating and measuring the effects of LD is challenging, so the equal importance is better than other methods? Further discussion on the applicability of LMI in other regions or research, as well as the limitations of this method, may be necessary.

Response 3

Yes, we completely agree with you that it is very complicated (if not impossible) to determine the exact effect of each process in the complex land multi-degradation mechanism. That's why, in these conditions of complexity and considering the essential objective of our approach (to quantify the number of co-

occurring processes and the types of their combination), we gave equal importance to all degradative factors, which is the best choice in the context of our proposed method/model (LMI).

Therefore, in accordance with your viable observation, in the manuscript there is some brief information related to this possible limitation of our methodology (“*Data quality and limitations*” section): “*Some potential limitations may exist in our methodological approach. These may emerge from ..., from assigning equal contributions to all layers in computing LMI (4), or from ... In the fourth case (4), we attributed equal importance to all drivers of degradation, although in reality some processes (e.g. soil salinization) could trigger more severe effects in the decrease of agricultural productivity, compared to other degradative factors (e.g. vegetation degradation). However, we avoided a weighting of the factors in modelling LMI, considering that, in principle, each process can lead to multiple negative effects for land productivity (Table 2), and especially that it is almost impossible to determine the exact impact of each process in the land multi-degradation mechanism.*”.

Also, following your suggestion, in the manuscript discussions we emphasized (very briefly, but clearly) the potential applicability of our model in other regions of the planet: “*Therefore, we consider the LMI proposed here can be a valuable interdisciplinary tool for the complex scientific assessment of LD, which is applicable at any spatial scale and in any region of the planet, if multiple and optimal environmental data are available.*”.

Comment 4

3. This MS has introduced a clear spatial pattern of multi-LD process. However, what factors contribute to such a pattern in Europe? For example, why large parts of the agricultural areas are exposed to one, two and three drivers of LD (page 4, the second paragraph from bottom). Additionally, an overview of the drivers of interacting LD process across the continent could provide decision-makers with more specific information. An introduction to the drivers of LD in the study region would be beneficial.

Response 4

Your observation is interesting and very useful, but we addressed this exact issue in the initial Figure 3, or in Figures 3 and 4 of this revised version of the paper. The driving forces of agricultural land degradation can be considered LD processes themselves, the individual pattern of which was clearly spatialized across Europe, in Figure 1 (section titled “*The continental picture of individual land degradation processes*”). Moreover, the pattern of spatial interaction (combination) between LD processes is addressed and mapped in Figures 3 and 4 (section titled “*Types of spatially interacting processes across the continent*”), while the detailed national statistics of major LD process associations (combinations) are provided in Supplementary Figures 1–3 and Supplementary Tables 3–12.

Thus, assuming that we understand your observation correctly, we have already addressed the driving forces of agricultural land degradation in Europe, by analyzing the individual degradative factors (LD processes themselves, Figure 1) and the interaction between these factors (the major types of LD combinations – Figures 3, 4 and Supplementary Tables, as previously mentioned). However, considering that the information on the interacting LD processes is indeed essential for decision-makers, we briefly included some additional mentions in this respect, in the “*Policy implications*” section: “*Under these circumstances, we call on the European Commission to consider our complex findings on co-occurring and interacting LD processes in Europe, in order to more effectively stabilize and mitigate land multi-degradation, and*

ultimately to achieve a land degradation-neutral continent in the coming years. Also, through the consistent and unprecedented examination of multiple drivers of degradation across 40 countries, our work presents novel insights beyond European policies.”.

Comment 5

Some minor modifications require further checking.

1. Why did author evaluate multi-LD processes only for agricultural land?

Response 5

For three reasons: 1) agricultural landscapes are critical for food production, but are, at the same time, the most vulnerable (in general) to degradative conditions, 2) some land degradation processes are specific to agricultural lands (e.g. soil nutrient imbalances, soil pollution via pesticides), and 3) agricultural lands are essential for many environmental (agricultural, climate, sustainable development) policies.

We highlighted once again (very briefly) the importance of agricultural environments in our study, in the last two paragraphs of the introduction: *“We focused the entire analysis on continental (pan-European) agricultural environments, which are critically important for food production, but generally highly vulnerable to multi-degradation. Consequently, here we integrated a complex set of LD processes that are strategically important to continental agricultural productivity, thus trying to provide a solid scientific basis for a more realistic and efficient implementation of LD-related policies across Europe.*

From our perspective, this multi-process modelling approach provides the most comprehensive and accurate assessment of LD in Europe, which can be crucial for applying various agricultural (e.g. Common Agricultural Policy)²⁷, climate (the European Green Deal)²⁸ and sustainable development (Sustainable Development Goals – SDGs)¹¹ policies on the continent.”.

However, the reasons for selecting agricultural landscapes in our research can be found in other sections of the article – for example in sections:

1) *“Data selection”*:

“In order to investigate land multi-degradation in Europe, we selected twelve processes that are highly representative for agricultural environments (Table 2). The twelve processes are the most relevant for highlighting the agricultural landscapes' degradation in Europe (and worldwide)², considering certain biophysical mechanisms, general or particular, that trigger various negative effects in land productivity (Table 2).”;

2) *“Final data modelling”*:

“LMI has been investigated at the level of all agricultural classes, but also strictly within arable lands, to emphasize the co-occurrence pattern of processes in the most important agricultural environments for European food security.”;

3) *“Land multi-degradation pattern in Europe”*:

“This classification was explored across the entire European agricultural land area (Fig. 2a), and a special focus was given to the arable lands (Fig. 2d), which are highly relevant for ensuring crop production and food security in Europe.”.

Comment 6

2. Page 2, second paragraph, should the full name be indicated when an institutional acronym is mentioned for the first time? Like IPCC.

Response 6

Considering this helpful observation, we included the name of the reports in the manuscript. However, we removed the acronyms from the text because we did not use them in other parts of the article, but we kept them in the References section, because some scientific papers use their acronyms to cite these reports.

Comment 7

3. In Table 3, the time period varies for each LD process. For example, some indicators are available for only one year (e.g., water erosion, 2023), while others have dynamic datasets (wind erosion, 2001-2021). How did the authors standardize the data time? Did you calculate the mean values for wind erosion by using 2001-2021 data?

Response 7

Indeed, for wind erosion we used mean values for the 2001–2021 period. For other degradative processes we used geospatial data from a single year (e.g. water erosion, with data representative for 2012), according to the databases available in the international literature.

We agree with you that it would have been ideal to have a standard time frame for all land degradation layers, but this is currently impossible as some datasets are available only for one year, while others are available for several years. In this context, we chose to use the available data for collecting / modelling each degradation process in Europe – for one year or, in the case of multitemporal existing data, we used mean multiannual values (instead of a single year value), considering that these data most likely reflect a more accurate picture of land degradation.

Your good observation was however addressed in the “*Data quality and limitations*” section, where we clearly stated this technical shortcoming of the data, which it was not possible to avoid: “*Some potential limitations may exist in our methodological approach. These may emerge from the different spatial/temporal resolution (1) and...; In the first case (1), it would have been ideal to use datasets with similar/identical spatial and temporal resolutions, which was however impossible, given the high number of data layers used, with different technical (pixel size and time periods) characteristics available in literature (Table 3).*”.

REVIEWERS' COMMENTS

Reviewer #1 (Remarks to the Author):

Dear authors,

I really appreciate the additional work put into applying a Random Forest model for estimating the model uncertainty. I believe this was a critical point to address and I am glad that the authors succeeded in providing now robust LD estimates with associated uncertainties.

I also believe that it was a very fine decision from the authors to better detail and emphasize the policy framework that constraints some of the analyses and methodological choices (e.g., thresholds). This is a crucial aspect of science to policy making, and therefore there is a need for better acknowledgment by the scientific community. I hope this paper will contribute to the matter.

My other concerns and remarks have also been addressed adequately, and text and figures altered in an adequate manner. I do not have additional comments.

Thank you for the thorough work!

Reviewer #3 (Remarks to the Author):

No more comments for this round.